# Semi-Supervised Diseased Detection from Speech Dialogues with Multi-Level Data Modeling

## Abstract

Detecting medical conditions from speech acoustics is fundamentally a weakly-supervised learning problem: a single, often noisy, session-level label must be linked to nuanced patterns within a long, complex audio recording. This task is further hampered by severe data scarcity and the subjective nature of clinical annotations. While semi-supervised learning (SSL) offers a viable path to leverage unlabeled data, existing audio methods often fail to address the core challenge that pathological traits are not uniformly expressed in a patient's speech. We propose a novel, audio-only SSL framework that explicitly models this hierarchy by jointly learning from frame-level, segment-level, and session-level representations within unsegmented clinical dialogues. Our end-to-end approach dynamically aggregates these multi-granularity features and generates high-quality pseudo-labels to efficiently utilize unlabeled data. Extensive experiments show the framework is model-agnostic, robust across languages and conditions, and highly data-efficient—achieving, for instance, 90% of fully-supervised performance using only 11 labeled samples. This work provides a principled approach to learning from weak, far-end supervision in medical speech analysis. The code is available at `https://anonymous.4open.science/r/semi_pathological-93F8`.

## 1 Introduction

The use of speech acoustics as a biomarker for disease detection presents a compelling yet challenging machine learning problem (Strimbu & Tavel, 2010; Califf, 2018). The core task is to learn a function that maps a raw audio signal, which is a complex, high-dimensional time series, to a clinical label. However, this problem is characterized by several fundamental constraints that complicate standard supervised learning approaches. First, the field is plagued by severe data scarcity. Annotating medical speech data requires costly expert knowledge from clinicians, making large-scale dataset collection difficult (Niu et al., 2023; Koops et al., 2023; Wu et al., 2023a). Second, the labels themselves are often inherently noisy. Clinical ratings, such as depression severity scores, can suffer from significant inter-rater subjectivity, meaning the supervision signal is not a ground-truth value but a noisy human assessment (Berisha & Liss, 2024).

The most distinctive challenge is the problem of weak, far-end supervision. In a typical screening scenario, a single label (e.g., "depressed" or "not depressed") is provided for an entire multi-turn conversation. This session-level label is the only direct supervision signal. However, to model the conversation, the audio must be processed into a sequence of fine-grained representations (e.g., at the frame or clip level). A critical modeling assumption is that the pathological state is not uniformly expressed throughout the session; a patient may not reveal symptomatic speech patterns in every line of response. Thus, the model must learn to identify the most salient, discriminative segments within a long sequence that led to the overall clinical assessment, without any direct segment-level guidance (Zolnoori et al., 2023; Agbavor & Liang, 2022; Martínez-Nicolás et al., 2021).

Existing methods often sidestep this granularity issue by segmenting long recordings and treating each segment as an independent sample (Wu et al., 2023b; Cheong et al., 2025; Li et al., 2025a), implicitly assuming uniform expression of symptoms—an assumption that is frequently invalid (Li

et al., 2025b; Han et al., 2023). Furthermore, the significant domain shift between general speech tasks and clinical applications hinders the direct transfer of existing semi-supervised learning frameworks (Diao et al., 2023; Park et al., 2020).

To address these core machine learning challenges, we propose a novel semi-supervised framework designed for audio-based medical detection. Our approach explicitly models the hierarchy of information in a clinical conversation: from frame-level acoustics to clip-level utterances to the final session-level diagnosis. We introduce a method to dynamically aggregate and weight these multi-granularity representations to match the far-end supervision signal, effectively learning to pinpoint critical segments within a session. By leveraging unlabeled data and explicitly modeling the sparse nature of symptomatic expressions, our method achieves robust performance even with extremely limited and noisy labeled data.

Experiments on two datasets incorporating depression and Alzheimer's detection demonstrate that with only approximately 11 labeled samples, our method can achieve 90% of the performance attained using the full training dataset. We further validate its effectiveness across diverse languages, medical conditions, and speech encoders, showing it matches fully-supervised performance using only 30% of the labels. A key feature of our method is its ability to dynamically generate high-quality pseudo-labels during training, efficiently leveraging unlabeled data without additional inference cost. This design enhances robustness, facilitates cross-lingual application, and aligns closely with real-world clinical scenarios. The main contributions of our work are as follows:

- We propose a novel, audio-only, model-agnostic semi-supervised learning framework for medical diagnosis from spoken dialogues. This framework is capable of simultaneously modeling data at multiple granularities, thereby enabling more comprehensive data utilization.

- We introduce a single-stage, end-to-end semi-supervised training method based on this framework. This approach processes complete long-form audio dialogues in a single pass and performs online updates of pseudo-labels to better leverage unlabeled data, all without incurring additional inference cost.

- We validate the effectiveness of our method across diverse languages, medical conditions, and speech encoder models. Our experiments demonstrate that with only 30% of the labeled data, our approach achieves performance comparable to its fully supervised counterpart trained on 100% of the data.

## 2 PROBLEM STATEMENT

This section provides a formal definition of the semi-supervised pathology detection task for speech-based clinical dialogues (illustrated in Figure 1). We begin by outlining the core formulation and the primary challenges inherent to this learning paradigm.

The problem is initially formulated as a $C$ class semi-supervised classification problem, where one class represents healthy participants and the remaining $C-1$ classes correspond to specific pathological conditions.

The labeled and unlabeled datasets are denoted as $\mathcal{D}_L = \{\mathbf{x}_i^l, \mathbf{y}_i^l\}_{i=1}^{N_L}$ and $\mathcal{D}_U = \{\mathbf{x}_i^u\}_{i=1}^{N_U}$, respectively, where both $x_i^l \in R^{t_i \times d}$ and $x_i^u \in R^{t_i \times d}$ are speech-based clinical dialogue samples of varying lengths (Chen et al., 2023a). Generally, the duration of each sample varies and exhibits significant variance. $N_L$ and $N_U$ represent the number of samples in the labeled and unlabeled data, respectively. The term $y \in \{0, 1, 2, ..., C-1\}$ is the one-hot ground truth label, which is exclusively available for the labeled data and indicates the class of the sample.

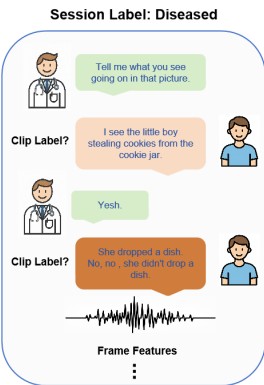

Figure 1: Pathological speech detection in clinical diagnostic dialogues.

For each sample $x_i \in \{D_L, D_U\}$ in the labeled and unlabeled data:

$$x_i = \{R_{i,1}, R_{i,2}, R_{i,3}, ..., R_{i,n}\} \tag{1}$$

$$R_{i,j} = \left\{W_1^{S_1}, W_2^{S_2}, W_3^{S_3}, ...W_m^{S_m}\right\} \tag{2}$$

Here, $R_{i,j}$ denotes the $j-th$ dialogue session in the $i-th$ sample, and $W_k^{S_k}$ represents the $k-th$ utterance from speaker $S_k \in \{INV, PAR\}$ within that session. The speakers consisted of investigators and participants. The variable $n$ is the total number of dialogue sessions in the sample $x_i$, while $m$ is the number of utterances in a given session. Notably, the utterances within each dialogue session are sequentially ordered. In contrast, different dialogue sessions are mutually independent, and thus no specific order is maintained among them. This formulation presents several fundamental challenges:

- **Data Scarcity and Noisy Labels:** The labeled set $\mathcal{D}_L$ is typically small due to the high cost of clinical annotations. Furthermore, the labels themselves often exhibit significant noise and subjectivity due to inter-rater variance in clinical assessments.

- **Session-Level Supervision with Sparse Manifestations:** Each dialogue session $x_i$ receives only a single global label $y_i$, despite consisting of thousands of acoustic frames and multiple conversational turns. Critically, pathological speech patterns may be *sparsely distributed* throughout the session—patients do not necessarily exhibit disease markers in every utterance or response.

- **Granularity Mismatch:** The supervision signal operates at the session level, while meaningful acoustic features must be extracted at much finer temporal resolutions (frame-level or clip-level). The model must therefore learn to identify which specific segments within a long dialogue are most indicative of the overall pathological condition, without explicit segment-level guidance.

These challenges necessitate a learning framework that can handle weak, far-end supervision while effectively leveraging unlabeled data to overcome annotation scarcity.

## 3 METHODS

We propose a novel semi-supervised learning framework that hierarchically models speech data at three distinct granularities: session, clip, and frame levels (Figure 2). At the session-level, which constitutes the main pipeline of our framework, the model is designed to process the entire audio sample $x$. We adopt an architecture commonly used in instance learning, where each utterance $W_k^{S_k}$ is encoded individually. Subsequently, a multi-head attention mechanism is employed to aggregate the features of each utterance. The resulting representation is then fed into a downstream detection task to yield the final result.

At the clip-level, the model trained in the main pipeline is leveraged to generate pseudo-labels for each utterance $W_k^{S_k}$. These pseudo-labels are then used to further train the audio encoder. This process enables the model to effectively capture the characteristics of each utterance in the dialogue, thereby facilitating the learning of sentence-level features by the audio encoder.

At the frame-level, we apply a Siamese network paradigm. By employing a contrastive loss, the model is trained to perform finer-grained modeling of frame-level features.

### 3.1 SESSION-LEVEL

The primary workflow of our framework, highlighted in green in Figure 2, processes an entire sample $x$ to produce the final detection result. To address the memory constraints of loading a complete sample at once, we partition each sample $x_i = \{clip_1, clip_2, clip_3, ..., clip_n\}$ into $n$ clips. Each $clip_i \in R^{t_i}$ is a vector sequence with a temporal length of $t_i$. Each $clip_i$ is then individually fed into an audio encoder $E$, to generate a corresponding embedding, $embed_i \in R^{t_i \times d}$. The resulting

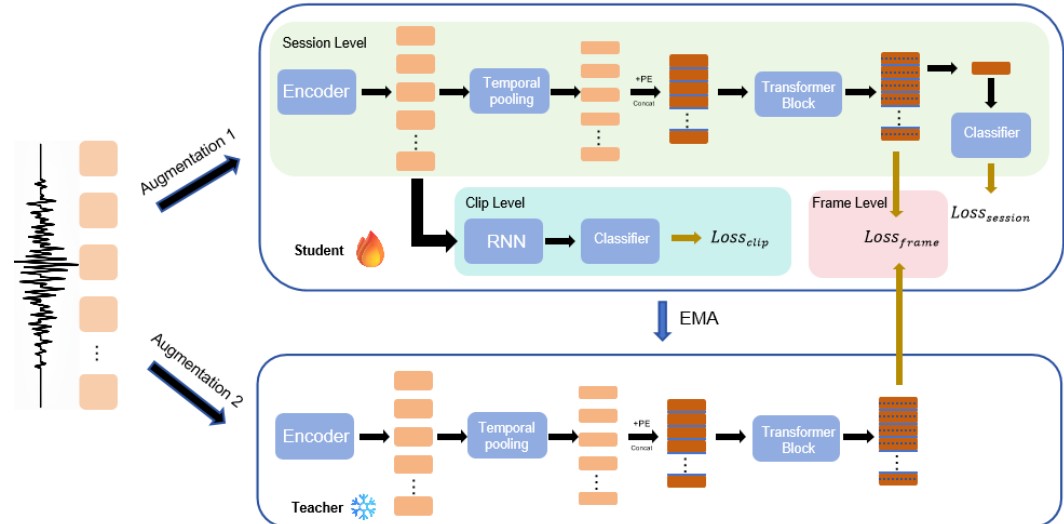

Figure 2: Model architecture overview. Our framework operates at three hierarchical levels: session-level (global dialogue representation via Transformer), clip-level (utterance-level modeling via RNN with pseudo-labels), and frame-level (acoustic feature consistency via MSE loss). The teacher-student framework with Exponential Moving Average (EMA) updates enables dynamic pseudo-label refinement during training.

embeddings may have varying temporal lengths $t$ but share a constant feature dimension $d$. For the audio encoder, standard architectures such as wav2vec2 (Baevski et al., 2020), HuBERT (Hsu et al., 2021), or WavLM (Chen et al., 2022) can be employed. To further reduce the data scale, the embedding $embed_i$ may optionally be passed through a temporal pooling layer, yielding a more compact representation, $embed_i$. Note that this step only reduces the temporal dimension $t$, while the feature dimension $d$ is preserved.

Following these steps, we obtain an encoded representation for each clip:

$$embed_{clip_i} = POOL(E(clip_i)) \qquad (3)$$

Subsequently, learnable positional encodings are added to the clip-level embeddings. The clip-level embeddings $embed_{clip}$ are first concatenated in their original sequential order to form a session-level embedding. This embedding is subsequently fed into a multi-layer transformer (Vaswani et al., 2017) to produce the final session-level representation $embed_{audio} \in R^{t \times d}$. Building upon this representation, a sample-level embedding is derived by aggregating features along the temporal dimension. This can be accomplished either by adding an additional layer to model global information or by employing a temporal attention mechanism for fusion. Finally, for the specific downstream task, a simple classification head is appended to the sample-level embedding to predict the final labels. During this process, pseudo-labels are generated for unlabeled data and incorporated into the training set. The model is then trained for the detection task in a supervised manner using the cross-entropy loss function.

## 3.2 CLIP-LEVEL

The objective of this method is to fine-tune the session-level encoder $E$, enabling it to model finer-grained data at the clip-level. This process is illustrated by the blue-highlighted portion of Figure 2. Specifically, the embeddings $embed_{clip}$ obtained from the session-level, optionally after a pooling operation, are fed as a sequence into a Recurrent Neural Network (RNN). Since a clip-level segment typically corresponds to a short sequence, such as a single utterance or a fixed-duration speech segment, standard RNN architectures like Gated Recurrent Units (Chung et al., 2014) or Long Short-Term Memory (Hochreiter & Schmidhuber, 1997) are employed to process this sequence and generate the final clip-level embedding:

$$embed_{clip_i} = RNN(clip_i) \tag{4}$$

The clip-level pseudo-labels are obtained directly from the main session-level pipeline and are subsequently used to supervise the training via a cross-entropy loss function. In contrast to prior work, this approach avoids the strong assumption that every utterance from a patient exhibits pathological features, while every utterance from a healthy individual is devoid of them. Moreover, this method can be trained on data containing dialogue from both investigators and participants, without requiring the explicit extraction of the participant' utterances.

### 3.3 FRAME-LEVEL

The objective of this method is to enable the model to capture finer-grained features at the frame level. To this end, we employ a siamese network paradigm, which consists of a student and a teacher model that share an identical architecture (The region highlighted in red in Figure 2.). The parameters of the teacher network are updated as an Exponential Moving Average (EMA) of the student network's parameters:

$$\theta_{teacher} \leftarrow m \cdot \theta_{teacher} + (1 - m) \cdot \theta_{student} \tag{5}$$

Throughout the training process, the teacher network remains frozen; no gradients are backpropagated through it, and only the student network is trained. For a given input $x$, we generate two distinct views by applying different data augmentations (one of which may be the identity transformation). These views are then fed into the teacher and student networks, respectively. The augmentation strategy employs common audio techniques such as speed perturbation, pitch shifting, and time masking. After being processed by the session-level pipeline of each network, we obtain the embeddings $embed_{teacher},\ embed_{student} \in R^{t \times d}$. Since these embeddings originate from the same sample, the objective is to enforce consistency between them. The loss function is therefore defined as:

$$Loss_{frame} = MSELoss(embed_{teacher}, embed_{student}) \tag{6}$$

### 3.4 ONLINE SINGLE-STAGE TRAINING RECIPE

In contrast to conventional multi-stage semi-supervised learning methods, our approach operates in a single stage, facilitating the online update of pseudo-labels. During the training process, after an initial warm-up period of $k_0$ steps, all pseudo-labels at both the audio- and clip-levels are re-evaluated and updated every $k$ steps. This update mechanism employs a threshold-based strategy: unlabeled samples with model-predicted confidence scores exceeding a predefined threshold are incorporated into the training set for the subsequent $k$ steps. Conversely, samples with scores below the threshold are excluded from (or optionally retained in) the training set for this duration.

In summary, the total loss for each training iteration is computed as a weighted sum of three distinct level-specific losses. The parameters of the teacher model are subsequently updated using the parameters of the student model. The loss function is defined as:

$$Loss = \alpha Loss_{session} + \beta Loss_{clip} + \gamma Loss_{frame} \tag{7}$$

where $\alpha$, $\beta$, and $\gamma$ are the weighting coefficients for the respective loss components.

## 4 EXPERIMENTAL SETUP

To validate the efficacy of our proposed method, we conducted experiments targeting two distinct pathological conditions: ***Depression*** and ***Alzheimer's disease***, using publicly available datasets in different languages. We evaluated our method under semi-supervised settings with varying proportions of labeled data, employing the Macro F1 Score to address class imbalance. Comprehensive ablation studies were performed to analyze the contribution of each component, and we compare

our approach against relevant baselines despite the limited prior work in audio-only semi-supervised pathology detection.

**Datasets**   To ensure fair comparison with prior work, we followed the same evaluation protocols established in the original dataset publications. Detailed dataset statistics, preprocessing steps, and experimental configurations are provided in Appendix A.

- **Depression Detection:** A Chinese EATD-Corpus dataset (Shen et al., 2022) of 162 participants (30 depressed), with 3-fold cross-validation.
- **Alzheimer's Detection:** An English ADReSSo21 dataset (Luz et al., 2021) with standard train/test splits. The dataset comprises a total of 237 samples, including 122 positive samples.

**Evaluation Protocol**   We adopted the Macro $F1$ Score as our primary metric to mitigate the effects of class imbalance, particularly relevant for the depression detection task. For the semi-supervised evaluation, we conducted experiments using varying proportions of labeled data (10%, 20%, 30%, 40%, 50%, and 100% of training samples) to comprehensively assess our method's efficiency.

**Training Details**   We employed the Adam optimizer with an initial learning rate of 2e-6 and a weight decay of 1e-8. The batch sizes for both labeled and unlabeled data were set to 2, with 4 gradient accumulation steps. The primary data augmentation methods included speed perturbation and time masking. The decay rate for the Exponential Moving Average (EMA) was set to 0.999. For the model architecture, features were extracted from the 10th layer of all audio encoders. The temporal pooling kernel size was set to 5. The transformer model comprised 3 blocks with 16 attention heads, while the RNN model consisted of a 2-layer bidirectional LSTM. Notably, as the EATD-Corpus is a Chinese dataset, we used HuBERT and wav2vec2 models pre-trained on Chinese speech datasets. Due to the unavailability of a WavLM model pre-trained on Chinese data, the version we employed was pre-trained on an English dataset.

## 5  RESULTS AND ANALYSIS

We evaluate our method's performance under varying proportions of labeled data against a session-level baseline that excludes pseudo-labeling. The results demonstrate our framework's effectiveness across both depression and Alzheimer's detection tasks, as shown in Table 1.

### 5.1  SEMI-SUPERVISED AND FULLY-SUPERVISED SETTING RESULTS

**Data Efficiency.**   Our method exhibits remarkable data efficiency, achieving strong performance with limited labeled data. For depression detection, our approach attained 90% of the fully-supervised baseline's performance using only 10% of the labels. Notably, with just 30% of the labels, it nearly matched the baseline's performance when trained on the full dataset. Similarly, for Alzheimer's detection, our method approached full-supervised performance with 30% of the data and surpassed it with only 40%.

**Performance Gains.**   Significant improvements over the baseline were observed across all label proportions. In depression detection, a notable gain of 4.59% was achieved at the 50% label ratio. The most substantial improvement for Alzheimer's detection was a 4.38% increase at the challenging 10% label ratio, highlighting the method's effectiveness in extremely low-data regimes.

**Full Supervision Enhancement.**   Crucially, our method outperformed the baseline even in the fully supervised setting (100% labels) for both disorders. This indicates that the integrated clip-level and frame-level components provide substantial performance benefits beyond pseudo-labeling, enhancing feature learning and representation robustness. These results collectively underscore the dual advantage of our framework: effectively leveraging unlabeled data to mitigate annotation scarcity while simultaneously enriching the model's representational capacity through multi-granularity analysis.

Table 1: Performance comparison of our method versus the baseline across different labeled data proportions for depression and Alzheimer's detection. Results report Macro F1 scores (standard deviation over 3 runs) under varying supervision levels (10%-100% of labeled data).

| Method | 100% | 50% | 40% | 30% | 20% | 10% |
|--------|------|-----|-----|-----|-----|-----|
| **Depression Detection** | | | | | | |
| Baseline | 59.53(1.51) | 57.41(5.48) | 55.78(7.14) | 56.04(7.98) | 55.00(7.25) | 51.73(2.39) |
| **Ours** | **63.26(1.34)** | **62.00(5.39)** | **58.51(9.00)** | **58.59(9.16)** | **57.70(9.07)** | **54.37(3.79)** |
| **Alzheimer's Detection** | | | | | | |
| Baseline | 71.25(1.42) | 70.18(1.12) | 69.80(1.47) | 67.79(1.49) | 67.45(0.56) | 65.09(0.55) |
| **Ours** | **73.01(0.60)** | **71.35(0.60)** | **72.14(1.46)** | **70.11(0.52)** | **69.80(0.56)** | **69.47(0.62)** |

**Comparisons with existing works.** Additionally, we compare our method's fully-supervised performance against existing approaches to establish its competitiveness. As shown in Table 2 and Table 3, our method achieves performance comparable to state-of-the-art methods on both depression and Alzheimer's detection tasks, despite not being specifically optimized for the fully-supervised setting.

Table 2: Comparison on depression detection. Methods marked with ∗ = reported in original publications.

| | F1 Score |
|--|----------|
| **CAMFM**∗(Xue et al., 2024) | 0.73 |
| **ACMA**∗(Iyortsuun et al., 2024) | 0.65 |
| **DepressGEN**∗(Liang et al., 2025) | 0.69 |
| **Ours** | 0.68 |

Table 3: Comparison on Alzheimer's detection. Methods marked with ∗ = reported in original publications.

| | F1 Score |
|--|----------|
| **Whisper-TL**∗Wu et al. (2024) | 0.77 |
| **CogniAlign**∗Ortiz-Perez et al. (2025) | 0.80 |
| **Wu et al. (2024)**∗ | 0.86 |
| **Ours** | 0.83 |

**Pseudo-label Analysis.** We analyze the evolution of pseudo-label quality throughout training in Figure 3, showing a consistent upward trend in the proportion of correctly labeled samples as the model converges. Our method can generate progressively higher-quality pseudo-labels. Notably, in later training stages, the pseudo-label accuracy for the 20% - 40% labeled data settings surpasses the performance of the one trained with 50% ground-truth labels (Table 1). This suggests that our framework effectively creates a self-improving training cycle where pseudo-labels eventually exceed the quality of additional manual annotations.

Furthermore, the frame-level component provides inherent robustness against pseudo-label noise. Since frame-level training operates independently of pseudo-labels and focuses on low-level acoustic patterns, it mitigates potential error propagation from incorrect session-level or clip-level pseudo-labels. This multi-granularity approach creates a balanced learning system where each component complements the others' limitations.

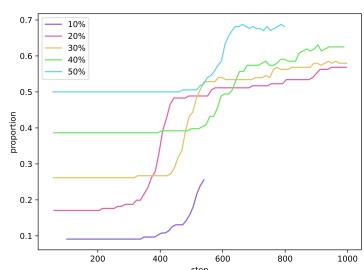

Figure 3: Evolution of pseudo-label accuracy during training on depression detection.

## 5.2 ABLATION STUDY

We conduct comprehensive ablation studies to validate the contributions of each component in our framework. The experiments are designed to address four key aspects: the efficacy of multi-granularity modeling, the impact of encoder trainability, the robustness to different audio encoders, and the handling of investigator speech.

**Multi-granularity Modeling Efficacy** Table 4 presents an incremental ablation of the three hierarchical components. Each level consistently contributes to performance improvements across all labeled data proportions, with the **frame-level component yielding the most significant gains**.

This demonstrates the importance of fine-grained acoustic analysis for pathological speech detection. In the fully-supervised setting (100% labels), where the session-level pseudo-labeling is inactive, the performance improvement validates the combined efficacy of clip-level and frame-level components.

Table 4: Ablation study of hierarchical components on depression detection. Results show Macro F1 scores (mean±std) with incremental addition of session-level, clip-level, and frame-level modules. "Ours-frozen" denotes training with frozen audio encoder.

|  | 100% | 50% | 40% | 30% | 20% | 10% |
|---|---|---|---|---|---|---|
| **Baseline** | 59.53(1.51) | 57.41(5.48) | 55.78(7.14) | 56.04(7.98) | 55.00(7.25) | 51.73(2.39) |
| **+Session-level** | - | 60.55(6.11) | 57.09(8.62) | 58.25(9.55) | 55.51(7.53) | 52.95(3.09) |
| **+Clip-level** | 60.78(4.52) | 58.30(5.41) | 56.11(8.34) | 56.55(8.27) | 55.52(7.75) | 52.50(2.73) |
| **+Frame-level** | 62.87(3.04) | 60.31(7.33) | 58.21(9.42) | 58.21(9.42) | 57.18(8.68) | 54.21(3.17) |
| **Ours-frozen** | - | 60.29(5.61) | 58.07(8.97) | **60.37(5.23)** | 56.96(5.90) | 52.82(2.45) |
| **Ours** | **63.26(1.34)** | **62.00(5.39)** | **58.51(9.00)** | 58.59(9.16) | **57.70(9.07)** | **54.37(3.79)** |

**Encoder Trainability and Component Isolation**   To further isolate each component's contribution, we conducted experiments with a frozen audio encoder. Under this condition, the clip-level component (which operates directly on encoder outputs) is effectively disabled. The maintained performance gain demonstrates the **joint effectiveness of session-level and frame-level components**. Notably, trainable encoders generally yield greater improvements, highlighting the importance of feature adaptation for medical speech tasks.

Table 5: Results of different audio encoders on depression detection

|  | 100% | 50% | 40% | 30% | 20% | 10% |
|---|---|---|---|---|---|---|
|  | | | wav2vec2 | | | |
| **Baseline** | 61.29(4.43) | 58.76(3.98) | 57.89(3.53) | 57.26(3.46) | 54.96(0.72) | 54.32(6.79) |
| **Ours** | **63.43(2.92)** | **59.75(3.80)** | **60.30(3.64)** | **57.85(4.06)** | **57.82(2.93)** | **55.45(2.54)** |
|  | | | WavLM | | | |
| **Baseline** | 59.80(3.72) | 56.98(6.63) | 57.85(5.25) | 58.31(4.16) | 59.37(4.32) | 53.37(7.59) |
| **Ours** | **63.56(4.97)** | **60.39(4.63)** | **60.73(5.89)** | **59.42(5.57)** | **60.54(7.15)** | **59.62(5.22)** |

**Architectural Robustness Across Encoders**   We evaluated our framework with three popular audio encoders: wav2vec2, HuBERT, and WavLM (Tables 5 and 6). Our method achieves consistent performance gains across all architectures, demonstrating its model-agnostic nature. An interesting observation emerges with WavLM on the Chinese EATD-Corpus, where performance degrades with more labeled data. We attribute this to cross-lingual transfer issues, as WavLM was primarily pre-trained on English speech, highlighting the importance of language-matched pre-training.

**Robustness to Investigator Speech**   As shown in Table 6, our method maintains performance improvements even when processing raw dialogues containing both participant and investigator speech. This eliminates the need for error-prone preprocessing steps like speaker diarization, making our framework more suitable for real-world clinical applications where clean speech segmentation is challenging.

Table 6: Different audio encoders and inclusion of investigators' speech segments on Alzheimer's Detection.

|  | 100% | 50% | 40% | 30% | 20% | 10% |
|---|---|---|---|---|---|---|
|  | | | wav2vec2 | | | |
| **Baseline** | 68.74(1.51) | 65.69(1.40) | 60.84(5.19) | 58.74(6.09) | 53.58(3.05) | 50.36(10.17) |
| **Ours** | **70.48(3.12)** | **65.91(0.83)** | **62.34(5.53)** | **64.51(2.29)** | **55.70(4.14)** | **53.61(3.01)** |
|  | | | with investigator | | | |
| **Baseline** | 72.84(0.54) | 72.68(1.41) | 70.50(1.09) | 68.21(2.20) | 66.25(1.63) | 66.93(0.47) |
| **Ours** | **73.62(0.56)** | **72.92(0.98)** | **72.44(1.07)** | **69.76(0.01)** | **70.42(1.01)** | **69.72(0.52)** |

## 6 RELATED WORK

Pathological speech analysis exhibits certain advantages in cross-lingual applicability and robustness to transcription errors. While multi-modal methods exist that combine acoustic, text, and visual information (Cheong et al., 2025; Thallinger et al., 2025; Wu et al., 2024), they face significant challenges, including error propagation from automatic speech recognition systems and limited generalization across languages and domains. Audio-only approaches offer a promising alternative by learning pathological patterns directly from acoustic signals. This is particularly valuable given the linguistic imbalance in available datasets, where most resources exist for high-resource languages like English and Chinese, while low-resource languages remain underserved. By bypassing linguistic content, these methods can achieve better cross-lingual transfer, making them more suitable for global healthcare applications.

However, current audio-only methods (Feng & Chaspari, 2024; Chen et al., 2023b; Zhou et al., 2022; Zhao et al., 2025) have predominantly focused on fully-supervised paradigms, typically employing transfer learning from general-purpose self-supervised audio models. The semi-supervised learning paradigm remains largely unexplored in this domain, despite its potential to address the critical challenge of limited labeled medical data. This gap is particularly notable given that semi-supervised techniques have shown success in other audio domains but face unique challenges in medical applications due to the sparse nature of pathological patterns in speech. Our work addresses this gap by proposing a novel semi-supervised framework specifically designed for audio-only pathological speech detection, leveraging multi-granularity analysis to effectively utilize both labeled and unlabeled data while maintaining cross-lingual applicability.

Semi-Supervised Learning (SSL) aims to enhance model performance by leveraging abundant unlabeled data. Prevailing methods are generally based on consistency regularization (Sohn et al., 2020), distribution alignment (Kim et al., 2020), and contrastive learning (Lee et al., 2022; Yang et al., 2022). Most of these methods focus on selecting reliable pseudo-labels throughout the training process (Gan et al.). However, the direct application of these methods to the medical domain is impeded by the multi-level and hierarchical nature of clinical dialogues. Furthermore, the reliance on far-end supervision poses additional challenges to improving the quality of pseudo-labels.

## 7 LIMITATIONS

Despite its strong performance, our method has limitations, primarily stemming from its nature as an audio-only approach. In contrast to multimodal systems, our model cannot leverage information from other modalities. However, this unimodal design offers distinct advantages. Modeling solely on acoustic information facilitates cross-domain generalization, reduces training data requirements, and results in a more parameter-efficient model. Furthermore, it obviates challenges inherent in multimodal approaches, such as potential modality conflicts.

## 8 CONCLUSION

In this work, we propose a novel, audio-only semi-supervised learning framework for medical diagnosis from speech-based clinical dialogues. Our method is uniquely designed to handle long-form medical consultation dialogues, simultaneously modeling speech data at three distinct granularity levels (session, segment, and frame) to ensure comprehensive data utilization. By dynamically generating high-quality pseudo-labels within a single-stage, end-to-end training process, our approach effectively leverages large volumes of unlabeled data without incurring additional inference costs. It avoids the limitations common in multi-modal methods. Our extensive experiments validate the effectiveness of the proposed framework. The efficacy of our method across diverse languages, medical conditions, and underlying speech encoders demonstrates its model-agnostic nature and strong generalization capability.

### ETHICS STATEMENT

The authors have read and adhere to the ICLR Code of Ethics. This work does not involve human subjects, identifiable private data, or harmful applications. All datasets used are publicly available

and were used in accordance with their original licenses and intended purposes. No external sponsorship or conflict of interest influenced the design or conclusions of this work.

## REPRODUCIBILITY STATEMENT

All code and source files are provided in the supplementary material and will be publicly released. Additional implementation details can be found in the training details section and `https://anonymous.4open.science/r/semi_pathological-93F8`.

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

## A  DATASET DETAILS

EATD-Corpus (Shen et al., 2022) is a publicly available Chinese depression dataset, which comprises audios and text transcripts extracted from the interviews of 162 volunteers. All the volunteers have signed informed consents and guarantee the authenticity of all the information provided. Each volunteer is required to answer three randomly selected questions and complete an SDS questionnaire. SDS is a commonly used questionnaire for psychologists to screen depressed individuals in practice (Shen et al., 2022). The EATD-Corpus consists of 162 samples, totaling 2.26 hours of audio data, which includes 132 samples from healthy controls and 30 from patients diagnosed with depression. To ensure a fair comparison with prior studies, we employed a 3-fold cross-validation scheme. We utilized only the audio data, partitioning all samples into three equal folds: two for training and one for testing. Furthermore, to maintain the stability of the results, we augmented the test set by reshuffling the data following the methodology of Shen et al. (2022), while the training set was kept unchanged. The final results are reported as the average over the three folds.

ADReSSo21 (Luz et al., 2021) is a publicly available English-language Alzheimer's Disease dataset, comprising two subsets for distinct sub-tasks. The dataset is balanced for age and gender and includes audio recordings from both investigators and participants. Each data contains recordings of a picture description task ("Cookie Theft" picture from the Boston Diagnostic Aphasia Exam). Those recordings have been acoustically enhanced (noise reduction through spectral subtraction) and normalized. To ensure a fair comparison with prior work, we utilized only the audio data and adhered to the Luz et al. (2021)'s splits for the training and test sets. A validation set was further partitioned from the training set, and the final results are reported on the test set. We conducted multiple experimental runs with different random seeds. Furthermore, as the majority of previous studies have focused on the first sub-task of ADReSSo21, namely the Alzheimer's Disease classification task, we also provide the results of our method on this sub-task for a direct comparison.

## B  TRAINING DETAILS

We employed the Adam optimizer with an initial learning rate of 2e-6 and a weight decay of 1e-8. The batch sizes for both labeled and unlabeled data were set to 2, with 4 gradient accumulation steps. The primary data augmentation methods included speed perturbation and time masking. The decay rate for the Exponential Moving Average (EMA) was set to 0.999. For the model architecture, features were extracted from the 10th layer of all audio encoders. The temporal pooling kernel size was set to 5. The transformer model comprised 3 blocks with 16 attention heads, while the RNN model consisted of a 2-layer bidirectional LSTM. Notably, as the EATD-Corpus is a Chinese dataset, we used HuBERT and wav2vec2 models pre-trained on Chinese speech datasets. Due to the unavailability of a WavLM model pre-trained on Chinese data, the version we employed was pre-trained on an English dataset.

## C  THE USE OF LARGE LANGUAGE MODELS (LLMS)

We disclose that we used Gemini-2.5-Pro to assist in polishing the language and improving the clarity of this paper. The model was used for grammar correction, sentence restructuring, and enhancing overall readability. All technical content, experimental design, results, and conclusions were authored and verified solely by the human authors. The LLM did not contribute to the generation of ideas, methods, or data analysis.

