# OpenReview forum: "Semi-Supervised Diseased Detection from Speech Dialogues with Multi-Level Data Modeling"
_ICLR.cc/2026/Conference — ICLR 2026 Conference Withdrawn Submission_

### Official Review · Reviewer_ZPiZ · 2025-10-28

**Soundness:** 2
**Presentation:** 2
**Contribution:** 1
**Rating:** 2
**Confidence:** 5

**Summary:**

The paper presents a semi-supervised framework for speech-based disease detection, focusing on Alzheimer’s and depression. It is clearly written and easy to follow, with well-organized sections and intuitive multi-level modeling (session-, clip-, and frame-level). The single-stage pseudo-label updating strategy is elegant and efficient, and the figures and tables are informative and well-presented.

However, the paper has several major weaknesses. The title and framing are misleading, suggesting a universal “speech disease detection” model without sufficient clinical grounding or literature coverage. The evaluation metrics (mainly F1) provide limited insight for healthcare relevance, and the work lacks explainability or clinical validation. The experimental analysis, while technically solid, offers little real-world or scientific interpretation, and the overall experimental scope is narrow.

Overall, while the paper is well-written and explores an interesting technical direction, it lacks depth in clinical understanding and healthcare relevance.

**Strengths:**

The paper is clearly written and easy to follow — it took only 2–5 minutes to grasp the main idea. It proposes an AI-based semi-supervised framework for disease detection from speech dialogues, focusing on binary classification for Alzheimer’s and depression. The organization is concise, with well-structured sections and a straightforward motivation. The idea of leveraging multi-level modeling (session-, clip-, and frame-level) to address weak supervision and limited labels is novel and intuitive. The paper also provides solid experiments across two datasets, demonstrating the model’s efficiency with limited labels and its robustness across languages and encoder architectures. Figures and tables are clear and informative, helping readers understand the hierarchical model structure and performance trends. The proposed single-stage pseudo-label updating strategy is elegant and contributes to better efficiency and stability compared to standard multi-stage SSL approaches.

**Weaknesses:**

This paper has several significant technical flaws and lacks important references.

* Title and Clinical Framing. The use of the title “Diseased Detection from Speech” is inappropriate. I am not sure whether the authors have any clinical background in speech and language disorders, but the detection of each speech-related condition (e.g., Parkinson’s, ALS, Aphasia) is fundamentally different. It is impossible to claim the development of a universal “speech disease” detection system without clinical validation. Such detection must be condition-dependent. In this paper, the generalization is purely based on the datasets used — Depression and Alzheimer’s. Therefore, the title should explicitly state “Depression and Alzheimer’s Detection” unless the method is proven robust across a broad range of disorders. I would also suggest the authors engage with clinicians or speech-language pathologists to better understand the clinical context. Additionally, the literature review is very limited. For example, citing only Strimbu & Tavel (2010) and Califf (2018) is insufficient to represent the entire field of speech-based disease detection.

* Evaluation Metrics. The choice of F1 score as the main evaluation metric is questionable. If the goal is to develop an AI model for healthcare applications, the evaluation should reflect clinical or human-centered alignment — for example, how well the AI model correlates with human expert scores. Accuracy and F1 alone provide limited insight. Speech patterns may overlap across conditions (e.g., a healthy individual with fatigue may exhibit similar patterns to early-stage patients). The current binary setup (two classes) oversimplifies the clinical landscape. If additional classes were introduced, the F1 score would fluctuate without offering meaningful interpretability.

* Lack of Explainability and Clinical Evidence. A critical weakness lies in the absence of interpretability or clinical validation. If the model classifies a sample as “Alzheimer’s,” what evidence supports that decision? How does the model justify its output? In medical AI, interpretability is essential. Without it, the system resembles a black box — a clinician would never accept a diagnosis simply because “the model predicted 80% Alzheimer’s.” This reflects a lack of fundamental understanding of AI in healthcare.

* Limited Experimental Insight. The experimental analysis, including ablations such as “Architectural Robustness Across Encoders,” provides minimal scientific or clinical insight. The results show incremental performance changes but offer no interpretation about what those differences imply for real-world or clinical deployment.

* Insufficient Experimentation. The overall scale of the experiments is quite limited. The work reads more like an exploratory report rather than a rigorously validated study.

Overall Assessment. The authors appear to lack basic knowledge of AI healthcare principles and should perform a much broader literature review before continuing this line of research. I give a score of 2, acknowledging that the team did conduct some experiments, which reflects a positive exploratory effort.

**Questions:**

What do you think is the core task in speech healthcare problems — specifically in disordered speech detection?

---

### Official Review · Reviewer_CgfG · 2025-10-31

**Soundness:** 2
**Presentation:** 2
**Contribution:** 2
**Rating:** 4
**Confidence:** 3

**Summary:**

This paper addresses the problem of semi-supervised disease classification from long clinical speech conversations, where only a single conversation-level label is available per sample. To overcome the mismatch between coarse supervision and fine-grained acoustic variability, the authors propose a multi-granularity learning framework that jointly models conversation-, segment-, and frame-level representations. Specifically, conversation-level embeddings provide global supervision; segment-level pseudo-labels are generated online through confidence-based updates; and frame-level features are regularized via EMA-based teacher–student consistency.
The framework aims to progressively propagate limited supervisory signals from coarse to fine temporal resolutions while maintaining temporal coherence. Experiments are conducted on two clinical speech datasets—EATD (Chinese depression detection) and ADReSSo21 (English Alzheimer’s detection)—using multiple pretrained speech encoders (WavLM, HuBERT, wav2vec2). Results show consistent improvements under partial-label settings and modest gains even under full supervision. Ablation studies confirm the benefit of combining multi-level objectives and online pseudo-labeling.

**Strengths:**

1.The paper addresses a well-motivated and practically relevant problem of semi-supervised learning from conversation-level labels in clinical speech analysis.
2.The multi-granularity framework (conversation, segment, frame) is conceptually clear and technically coherent, combining pseudo-labeling and EMA-based consistency in a single-stage pipeline.

**Weaknesses:**

1.While the hierarchical integration is elegant, the method remains conceptually close to existing semi-supervised frameworks such as FixMatch, Mean Teacher, and BYOL-like consistency learning. The absence of comparisons with such strong baselines weakens the claimed novelty. The improvement could partially stem from task-specific tuning rather than fundamentally new learning principles.
2.Key implementation details are under-specified, particularly regarding pseudo-label confidence computation, thresholding strategy, update frequency, and the weighting coefficients (α,β,γ) of the multi-level losses. These factors critically affect the stability and reproducibility of the results. The lack of sensitivity analysis or ablation makes it unclear whether the reported gains generalize across datasets or rely on delicate hyperparameter tuning.
3.The datasets are relatively small and imbalanced (e.g., only 162 cases in EATD). Reported standard deviations are large, yet no statistical significance tests are provided. Consequently, some performance gains might not be statistically reliable. Additionally, the generalization claims across languages and disorders are not fully substantiated by controlled cross-domain experiments.
4.The cross-lingual generalization claims are insufficiently supported. Although the study involves both Chinese (EATD) and English (ADReSSo21) datasets, each model is trained and tested within its own domain. The paper does not include any explicit cross-lingual transfer or domain-shift experiments, even though such evidence is central to the stated goal of building language-agnostic disease detection models.

**Questions:**

1. Could the authors specify how the pseudo-label confidence is computed (e.g., maximum class probability, temperature-scaled logits, entropy)? How are the update interval k and confidence threshold chosen or adapted across datasets and label ratios?
2. The training objective includes multiple loss terms weighted by (α,β,γ). How are these hyperparameters selected? Are they tuned jointly or fixed?
3. Since segment-level labels are derived from conversation-level predictions, what measures are taken to prevent error reinforcement during training?

---

### Official Review · Reviewer_PfDY · 2025-11-01

**Soundness:** 1
**Presentation:** 1
**Contribution:** 2
**Rating:** 2
**Confidence:** 4

**Summary:**

The paper presents an approach to detect medical conditions from speech by leveraging frame-level, segment-level, and session-level representations obtained from unsegmented clinical dialogues. The paper claims that the proposed approach is model-agnostic, robust across languages and conditions, and highly efficient when labeled data is scarce, reporting performance as high as 90%.

The paper points to a GitHub repo, however the repository is empty and provides no information about the implementation of the approach, but only points to the data resources that have been cited in the paper.

**Strengths:**

The paper addresses an important problem of having limited labeled data where training models that can generalize well is often difficult. While the paper is well motivated, and the problem statement is well defined, the strength of the paper does not provide sufficient details on the implementation.

**Weaknesses:**

The paper points to a GitHub repo, however the repository is empty and provides no information about the implementation of the approach, but only points to the data resources that have been cited in the paper.

There are several issues with the draft:
(1) The title has a typo: " .. Diseased Detection ... " it should be " .. Disease Detection.
(2) At multiple points in the paper, the statements were made as too general "... standard architectures such as ... can be used", "This can be accomplished either by adding an additional layer to model global information or by employing a temporal attention mechanism for fusion" ... these statements should be corrected and the paper should clearly state what was actually done in the work presented.

**Questions:**

(1) It is shown in equation (3) that pooling was performed, which pooling was used and why? Was it imperially decided or heuristically?

(2) "In contrast to prior work, this approach avoids the strong assumption that every utterance from a patient exhibits pathological features..." >> not clear how the assumption is tackled in this work. it will be useful to clearly elaborate how this assumption is addressed in this work.

(3) "We analyze the evolution of pseudo-label quality ..." >> it is not clear how the pseudo-labels were generated.

(4) "We evaluated our framework with three popular audio encoders: wav2vec2, HuBERT, and WavLM (Tables 5 and 6) .." Neither table 5 or 6 specify any result from HuBERT.

**Details Of Ethics Concerns:**

The paper seems to have some strange errors, starting with a typing error in title where it says "Diseased Detection"

At multiple points in the paper, the statements were made as too general "... standard architectures such as ... can be used", "This can be accomplished either by adding an additional layer to model global information or by employing a temporal attention mechanism for fusion" ... these statements should be corrected and the paper should clearly state what was actually done in the work presented.

The paper states: ""We evaluated our framework with three popular audio encoders: wav2vec2, HuBERT, and WavLM (Tables 5 and 6) .." Neither table 5 or 6 specify any result from HuBERT.

---

### Note · Authors · 2025-11-15

I have read and agree with the venue's withdrawal policy on behalf of myself and my co-authors.